# Using the Radial Distribution Function to Analyze Atomic Force Microscopy Images of Colloidal Systems

**DOI:** 10.3390/ijms26010210

**Published:** 2024-12-30

**Authors:** Sergey V. Kraevsky, Anastasia A. Valueva, Maria O. Ershova, Ivan D. Shumov, Irina A. Ivanova, Sergey L. Kanashenko, Ilya A. Ryazantsev, Yuri D. Ivanov, Tatyana O. Pleshakova

**Affiliations:** Institute of Biomedical Chemistry, Pogodinskaya Str., 10, Moscow 119121, Russiamotya00121997@mail.ru (M.O.E.); i.a.ivanova@bk.ru (I.A.I.); serkanash@mail.ru (S.L.K.); ilryazancev@yandex.ru (I.A.R.); yurii.ivanov.nata@gmail.com (Y.D.I.); topleshakova@yandex.ru (T.O.P.)

**Keywords:** atomic force microscopy (AFM), gold nanoparticle, horseradish peroxidase (HRP), radial distribution function (RDF), Poisson distribution, colloidal particles

## Abstract

Biomacromolecules generally exist and function in aqueous media. Is it possible to estimate the state and properties of molecules in an initial three-dimensional colloidal solution based on the structure properties of biomolecules adsorbed on the two-dimensional surface? Using atomic force microscopy to study nanosized objects requires their immobilization on a surface. Particles undergoing Brownian motion in a solution significantly reduce their velocity near the surface and become completely immobilized upon drying. Using radial distribution function (RDF) methods, it is possible to obtain information about the presence of short-range or long-range order in the arrangement of immobilized colloidal particles. In this work, RDF is applied to immobilized gold nanoparticles (AuNPs) and horseradish peroxidase molecules on mica. It is shown that AuNPs maintain mobility on the mica surface when water is present. Upon immobilization, AuNPs organize into an amorphous structure exhibiting short-range order. Protein molecules are immobilized randomly, and their surface density is well described by the Poisson distribution.

## 1. Introduction

Atomic force microscopy (AFM) is one of the most popular analysis tools used in modern scientific research. AFM is used both in material sciences and in the study of living systems in biological and medical sciences. In the study of living systems, a particular advantage of AFM compared to cryo-electron microscopy and X-ray methods is the ability to perform measurements under near-native conditions. This makes it possible to relate information about the structure or dynamic conformational changes to the biological function of the investigated object [1].

The immobilization of investigated objects on surfaces is a necessary condition for the application of AFM. If the investigated objects are nanoscale biomacromolecules (proteins; nucleic acids), the surface should be atomically smooth so that surface roughness (irregularities) does not interfere with the visualization and study of these objects. Mica (muscovite), highly orientated pyrolytic graphite, or silicon are traditionally used for AFM studies. These substrates are atomically smooth, and the advantage of mica and graphite can also include easy surface renewal by shearing off the top layer. Immobilization methods can provide both reversible binding to the surface (hereinafter the term “adsorption” is used) and irreversible binding through the formation of a covalent bond between the functional groups of the surface and the object under investigation. In [2], the authors utilize AFM for the immobilization of multiple probe molecules with micrometer resolution for biochip preparation. Two-dimensional (2D) arrays of neutravidin with biotinylated oligonucleotides were visualized by fluorescence microscopy.

Biomacromolecules generally exist and function in aqueous media. There is a need to estimate the state and properties of molecules in the initial solution (the three-dimensional (3D) system) from the properties of biomolecules sorbed on the surface (the 2D system). The functionality of the family of AFM methods provides a wide range of possibilities for 2D system investigation. However, in order to translate information about 2D systems to the properties of biomolecules in solution, a better understanding of the interaction of the investigated objects with the surface is necessary.

In the case of the adsorption of biomacromolecules, deposition from water–salt solutions is used; the concentration and pH of the solution; the temperature, and time of incubation on the surface; and the methods of washing the surface from unbound components are selected. For the task of studying the structural and functional characteristics of biomacromolecules, adsorption is the preferred immobilization option, which ensures the presence of objects on the surface, but at the same time, the impact on them is minimal. When interpreting the obtained results of AFM studies, it should be taken into account that biomacromolecules are sorbed on the surface, which makes a certain contribution to the properties of these molecules. For example, when determining the nanomechanical parameters of nanoscale objects, a rigid substrate can make a significant contribution [3]

Enzymes can maintain functionally active conformation while immobilized on surfaces. As shown in [4], the proper orientation of ferredoxin–NADP+ can be achieved by mica surface modification. The activity maintenance is clearly demonstrated by modifying the cantilever of its redox partner ferredoxin and the direct measurement of adhesion forces using single-molecule force spectroscopy.

If the task of an AFM study is to characterize the structural properties of a biomacromolecule—for example, the linear size of protein globules or DNA chains— the mutual arrangement of objects on the surface is not a significant factor and is not taken into account. The main requirement for objects of study in this case is that biomolecules should be sorbed as individual objects that can be visualized using AFM. However, researchers may have other tasks besides the study of the structural features of biomacromolecules.

Protein solutions can be considered colloidal systems in which one phase is represented by nanoparticles—biomacromolecules of the order of 10 nm in size—which are in the second liquid water–salt phase. According to calculations [5], the distance between biomolecules in a 1 M solution is 1.18 nm, and in 1180 nm in 1 nM. Decreasing the concentration by nine orders of magnitude leads to an increase in the distance between molecules by three orders of magnitude. It is also known that intermolecular interaction between biomolecules occurs at a distance of 2–3 nm in solution. Do these dependencies hold for molecules at the surface, and is there interaction at greater distances? This is one of the questions that needs to be answered to be able to translate information about a 2D system into the properties of a 3D system.

A significant contribution to the behavior of particles in colloidal systems is the solvent. If the solvent molecules can form hydrogen bonds (like water), such a three-dimensional hydrogen bonding network will determine the solvophobic interactions of the system components, i.e., such a redistribution of matter that minimizes the surface of neutrally or weakly charged particles [6,7]. Such interactions can exceed the Coulomb repulsion of homonymously charged particles. For example, the authors of [8] have shown that micron-sized silicon particles with negatively charged SiO^−^ and COO^−^ groups, exposed on the surface, can attract through water over a wide pH range.

When thin films of colloidal solutions are dried on a surface, the particles can redistribute due to the surface tension gradient (the Marangoni effect) [9].

As mentioned above, microscopy techniques (e.g., AFM and transmission electron microscopy (TEM)) have the potential to provide accurate measurements of the nanoparticle number concentration and number size distribution [9,10,11]. For AFM measurements to be representative, it is absolutely critical that the sample is firmly bound to the substrate surface [10].

At the immobilization stage, if the particles under study are negatively charged, they experience a repulsive force from the mica surface. So, for the immobilization of [11,12] negatively charged DNA molecules on mica, divalent magnesium cations are additionally used to sorb the objects successfully. The alternative modification is to graft positive aminopropyltriethoxysilane (APTES), poly-L-lysine (PLL), or spermidine (Spdn) on the mica surface. APTES-functionalized mica can bind DNA under a wide range of buffer conditions without the need for added magnesium ions [13,14,15]. The work of [16] represents approaches for the efficient immobilization of DNA origami on the PLL and Spdn-functionalized mica surface. PLL and Spdn coatings were found to exhibit the efficient immobilization of DNA origami from PBS buffer. In particular, the adsorption of DNA origami structures onto Spdn-modified mica surfaces resulted in significantly improved surface coverage compared to Mg^2+^-mediated adsorption on freshly cleaved mica, with approximately 70% of the adsorbed structures remaining intact. Modification with PLL showed similar results to the addition of Mg^2+^ in terms of surface coverage but allowed for the acquisition of higher-resolution AFM images, making this surface particularly promising for the structural characterization of objects. However, when immobilizing DNA origami from water on PLL and Spdn surfaces, the coverage of structures was reduced compared to Mg^2+^-mediated immobilization on freshly cleaved mica, and the proportion of damaged structures increased. In the case of citrate-encapsulated gold nanoparticles, mica surface modifiers are applied, either completely drying the drop without washing or using ultracentrifugation [17]. In this case, preference is given to the method of ultracentrifugation (for example, under the conditions of 150,000× *g* for 60 min) [17] because during immobilization from the drying drop, in AFM images, there are artifacts associated with a local increase in salt concentration. This is valid for the task of reliably determining particle sizes and concentrations.

AFM in combination with fishing—the concentration of biomolecules from a volume on a specially prepared surface—can be considered a method for determining small concentrations of biologically relevant molecules in solution [18,19]. To solve this problem, it is also important to understand the distribution of biomolecules on the substrate surface. If biomolecules are distributed according to a known mathematical law, then the recalculation of the number of objects on the surface into the volumetric concentration of the solution is possible with a given error based on the data from scanning only a part of the surface.

When nanoscale objects are deposited (adsorbed) from the volume onto the surface, their immobilization occurs, i.e., a significant decrease in the diffusion rate up to the complete stopping of the object. This is due to a reduction in degrees of freedom—the transition from 3D to 2D—and an increase in concentration and sorption properties of the surface. Such immobilization is similar to the phase transition of a gas into a liquid, whose structure can be described by a radial distribution function (RDF). In general, the use of RDFs constructed, for example, directly from optical microscope images [8,20,21] is useful for quantifying the thermodynamic characteristics of colloidal systems. This function serves as a measure of the orderliness of a particle’s 2D system. Thus, if the particles are randomly distributed and do not interact, the RDF has a characteristic appearance, an example of which is shown in Figure 1b. If the particles are mobile with long-range interactions, i.e., interactions at distances much larger than the particles themselves, the RDF will take the form characteristic of amorphous structures or liquids (Figure 1e). It is expected that factors such as the availability of adsorption centers on the surface and the properties of the deposited biomolecules will influence the value of adsorption density—the number of objects per unit area.

RDF can be calculated directly from optical or confocal microscopy data based on information about the particle position in the acquired images. However, this method is applicable only for particles available in size for optical detection (in the micrometer range). The advantage of using optical systems over AFM is the possibility of time resolution; that is, in addition to static information, it is possible to access the dynamic parameters of the system, for example, diffusion constants [22]. Spatial resolution at the level of micrometer-sized particles allows us to investigate the morphology and dynamics of local structures in glasses [23] gels [21], and the anisotropic suspension structure in pressure-driven flow [24]. An optical system cannot be used to study systems of individual biomacromolecules because the objects are several nanometers in size.

The arrangement of sorbed objects on the surface may occur randomly or obey some law. The answer to this question will help the researcher correctly interpret AFM visualization data. During adsorption, there are two main variants of the arrangement of objects on the surface. In the first case, molecules approach the surface by diffusion and are sorbed or not sorbed at the point of contact between the molecule and the surface, assuming the equivalence of the entire surface for adsorption. The second option is the assumption of the heterogeneity of surface properties and the presence of binding sites that are preferred for the adsorption of molecules. This is a variant of “decorating” the surface, reflecting the surface properties, and in this case, the influence of the substrate factor on the mutual arrangement of AFM-visualization objects becomes especially important.

For the interpretation of AFM research data (2D systems) in terms of characterizing the interaction of particles in solution (3D systems), the distribution of particles on the surface can be used, but only in the first adsorption case described above. For this purpose, it is necessary to ensure that the surface itself is neutral with respect to adsorbed particles; i.e., it is devoid of inhomogeneities that have an advantage in particle adsorption. Otherwise, the distribution will reproduce (decorrelate) these inhomogeneities (surface defects; preferred particle adsorption centers). In the simplest case of non-interacting particles, they are sorbed onto the surface randomly (Figure 1a). It is known that in such a case. The distribution density will correspond to a discrete Poisson distribution.
*P*(*n*) = λ^*n*^e^−λ^/*n*!,
where *P*(*n*) is the probability that there will be n particles in the chosen surface area, given that the average density (mathematical expectation) of particles in the same area is λ particles.

A peculiarity of the Poisson distribution is that its dispersion, *σ*^2^*poisson*, is equal to λ. Moreover, the properties of the distribution of particles do not depend on the size of the area on which these particles are located.

In order to make sure that the particles do not interact with each other and that their arrangement obeys the Poisson distribution, it is enough to select surface areas of a certain area, count the number of particles on each area, plot the distribution of the number of particles in the selected areas, and check whether the hypothesis of equality, *σ*^2^*experimental* = λ, is indeed fulfilled. Further, if there is no reason to reject this hypothesis, there is no reason to reject the assumption that the particles do not interact with each other. With this approach, an estimate of the density of particles per unit surface area can be made. For example, if we choose an area that contains 100 particles, the variance of *σ*^2^*poisson* will be 100, and the standard deviation will be √100 = 10; i.e., all other areas of the same area will contain (100 ± 10) particles. By dividing 100 by the value of the selected area, we can estimate the particle density (1/surf) with an accuracy of 10/100 = 0.1 or 10%.

If we choose an area with 10,000 particles, the standard deviation is √10,000 = 100. So, 100/10,000 = 0.01 or 1%. By dividing 10,000 particles by the area, we find the mathematical expectation of the particle density with an accuracy of 1%.

If the dispersion measured in the experiment turned out to be smaller, *σ*^2^*experimental* < λ, such a distribution is called sub-Poisson or quasi-periodic (an example is provided in Figure 1f). In this case, there will be a negative correlation of particle deposition on the surface: the appearance of one particle on the surface reduces the probability of the appearance of another particle in the immediate neighborhood. This behavior suggests a possible repulsion of particles; i.e., there is a region around each particle that is free of others. This can also be verified by plotting the RDF of particles on the surface.

When the inverse inequality *σ*^2^*experimental* > λ, we obtain a super-Poisson process with particles collected in clusters with large distances between these clusters, such clusters can be visualized well by AFM. The above-described cases of object locations on the surface, the corresponding RDFs, and the distribution density functions are shown in Figure 1. As can be seen from Figure 1c,f, dispersion is smaller for repulsive objects than for non-interacting (Poisson) ones.

The aim of the present work is to experimentally verify the above reasoning using the AFM technique for two-particle systems in solution, gold nanoparticles (AuNPs) and proteins.

When AuNPs with citrate shells are immobilized on the mica surface, the formation of a structure similar to amorphous bodies is simulated. Being negatively charged, AuNPs spread among themselves in the drop and from the surface of mica surfaces and, when carefully dried, form a characteristic pattern. The choice of AuNPs as test objects is due to their properties. AuNP suspension is a classical colloidal system (3D), which can be translated into a 2D system and investigated by AFM. The main parameter of AuNPs is size, which can be determined by various methods, including atomic force microscopy, spectrophotometry, and electron microscopy. The obtained data can be compared with each other, as AuNPs hardly change their size during measurement with different methods. In the present work, apart from AFM, spectrophotometry has been employed as a reference method for characterizing AuNPs in order to determine their size and estimate their aggregation state.

The disordered structure of object distribution on the surface, obeying the Poisson distribution, can be found in the adsorption of most proteins onto mica. In the present work, the horseradish peroxidase protein (HRP) is used as a model object to investigate the arrangement of objects on the surface. This protein is an enzyme of the oxyreductase class with a molecular mass of ~44 kDa (kg/mol), which has been widely studied for many years and has shown promise in the field of biotechnology [25] in a surface-immobilized form. It is known that HRPs can be adsorbed on mica both as monomers with heights of 1.0 ± 0.2 and as oligomers with heights exceeding 1.4 nm [26].

The study of the location of sorbed particles on surfaces is important not only in AFM studies. The immobilization of bio-objects on various surfaces is widely used in biosensor systems and biotechnology. By understanding the mechanisms occurring on the surface during the adsorption or covalent immobilization of biomolecules, the influence of the nature of the objects and the surface on the localization of molecules will improve the efficiency of various bioanalysis systems, bioproduction systems, and filters.

## 2. Results

### 2.1. AFM-Visualization of Horseradish Peroxidase Molecules

Figure 2 shows an example of the AFM image of objects on the mica surface after incubation with HRP solution (A) and a histogram of object height distribution (B). According to [27,28,29], the isoelectric point of HRP is approximately 7.2 to 7.8. Therefore, under PBSD buffer conditions with a pH of 7.4, the protein charge is expected to be neutral or slightly positive. The AFM visualization resulted in a series of images, typical examples of which are also shown in the Appendix A. In some cases (rather rare), aggregated structures of objects were visualized in the image (Appendix A), but this type of image was not analyzed in this work since protein aggregation can occur both in solution and on the surface, and this issue requires separate consideration. Most images were characterized by the presence of freestanding compact objects up to 2 nm in height (Figure 2a,b and Appendix A), which corresponds to the monomeric form of the protein, as we have shown previously [26]. The RDF of peroxidase molecules on the mica surface plotted for this AFM image is shown in Figure 2c; similar data were obtained for other images of this type (Appendix A). Also for this image, the histogram of the density distribution was plotted (Figure 2d). The statistical analysis showed that the density distribution corresponds to a Poisson distribution with 0.95 confidence.

### 2.2. Visualization of AuNPs by AFM

Figure 3 shows an example AFM image of objects on the mica surface after incubation in a AuNP solution (Figure 3a) and a histogram of the absolute height distribution of the objects (Figure 3b).

As with HRP imaging, the full gallery of AuNP images included rare images with aggregated structures (Appendix A), which are not the focus of this work. Most images were characterized by the presence of freestanding compact objects up to 15 nm high (Figure 3a,b and Appendix A), which were assigned to individual AuNPs based on spectrophotometric analysis data (see Section 2.3). The AuNP distribution function over the mica surface constructed for this AFM image is shown in Figure 3c; similar data were obtained for other images of this type (Appendix A). A distribution histogram was also plotted for this image (Figure 3d). The statistical analysis showed that the density distribution does not conform to the Poisson distribution with the same confidence as for HRP.

### 2.3. Characterization of AuNPs by Spectrophotometry

The absorbance spectrum of an aqueous AuNP suspension depends on both the size and concentration of AuNPs in the suspension [30,31,32]. Figure 4 displays typical absorbance spectra obtained in our experiments.

As can be seen from Figure 4a, for the undiluted AuNP suspension, the absorbance peak wavelength can be observed at 521 nm. This wavelength corresponds to a nanoparticle diameter of 16–17 nm [30,32]. According to [30], the concentration of AuNPs in the cuvette corresponds to 1.27 × 10^12^ particles/mL. This means that the AuNP concentration in the initial suspension is 2100/900 × 1.27 × 10^12^ = 2961.27 × 10^12^ particles/mL. The data obtained in the experiments on the evaluation of the effect of the dilution of the AuNP suspension on the aggregation of AuNPs are presented in Figure 4b. The spectrum shown in Figure 4b indicates a predictable decrease in the peak intensity at a 10-times-lower AuNP concentration. Given this, virtually no shift in the wavelength of the absorbance peak can be observed, indicating the absence of dilution-induced AuNP aggregation.

## 3. Discussion

Two colloidal systems (3D), a protein solution, and a suspension of gold nanoparticles were investigated in this work. For this study, the objects from the solutions were sorbed onto the surface (2D system) and visualized using AFM.

The classical spectrophotometric method was also used to characterize the 3D AuNP system. The wavelength corresponding to the absorbance maximum (peak) of AuNPs in a suspension depends on their size [30,31,32] (and shape), and the peak intensity is directly proportional to their concentration in the suspension [30]. In this respect, spectrophotometry represents a very convenient tool for the characterization of AuNP suspensions, as it allows one to perform simultaneous determinations of the concentration and size of the nanoparticles in a single measurement. For 10 nm diameter spherical AuNPs, the peak wavelength is typically between 515 and 520 nm [30,32], increasing to 570–572 nm for 100 nm AuNPs [30,31] owing to the so-called red shift effect [32]. According to He et al., an aqueous suspension of 12 nm AuNPs has an orange-red color, while a suspension of larger AuNPs (of the order of 40 nm) has a purple color [32]. This phenomenon allows one to reveal AuNP aggregation. AuNP aggregates have larger sizes and, hence, are characterized by longer wavelengths corresponding to the absorbance peak [32]. The results of spectrophotometric measurements (see Figure 4) indicate that the size of the AuNPs synthesized in the present work is 16–17 nm, and their concentration in the cuvette corresponds to 1.27 × 10^12^ particles/mL. It has been demonstrated that the dilution of the initial suspension to the concentration used in the AFM study does not lead to AuNP aggregation (see Figure 4b).

The size of AuNPs determined by AFM and EM (Appendix A) in a 2D system is 10–15 nm, which is consistent with the spectrophotometry data. According to the AFM visualization data (Figure 3b), the particles are arranged on the surface with a short order and do not approach each other at distances smaller than 0.5 μm (Figure 3c). The appearance of the radial distribution function shown in Figure 3c suggests that there are repulsive forces between the nanoparticles that prevent them from approaching a distance of less than 0.5 µm. However, the nature of these forces is not obvious. On the one hand, it is known that a AuNP suspension is an example of a colloidal system in which the particles interact with each other; since the particles are homonymous in charge, it is expected that their interaction is based on the principle of repulsion. On the other hand, also according to the repulsion principle, the interaction of AuNPs with the surface of negatively charged mica during adsorption should be observed. According to the obtained AFM visualization data, the interaction of particles can indeed be observed. However, the ordered arrangement of AuNPs on the surface can be caused by various factors: the presence of electrostatic repulsion forces [33], the ordering of the Al^3+^ arrangement in tetrahedral AlSi_3_ [34], the presence of repulsive forces between particles in solution with a hydrogen bonding network, and the surface tension when a drop dries [8].

The colloidal system of the HRP protein is not as well characterized in the literature as the AuNP system. For example, the size of the protein globule in the native state cannot be determined by spectrophotometry in analogy with AuNPs. Size determination by X-ray diffraction analysis is performed under conditions that do not correspond to the conditions of protein function. Using AFM, we previously estimated the height of the HRP protein globule sorbed on the surface [26]. In the present work, we obtained another piece of information about this colloidal system. The processing of AFM imaging data for HRP biomolecules on the surface suggests that there is no interaction between HRP biomolecules. Thus, according to the radial distribution function of objects on the mica surface (Figure 2c), the minimum distance between particles cannot be determined. The objects are randomly arranged on the mica (compare Figure 1b and Figure 2c). In this case, the distribution of objects on the surface should obey a Poisson distribution (see Figure 1c), which is confirmed by the results of processing the AFM visualization of HRP biomolecules presented in Figure 2d. The obtained data do not refute the hypothesis of the random distribution of peroxidase molecules in the 2D system.

The approach proposed in this work can be used for the study of nanoscale colloidal systems, which are important to study under conditions close to the conditions during the formation of these systems. For example, it is good when the solution–surface system under study is well known—for example, when the adsorption isotherm is known. Then, at equilibrium, one can judge the concentration (3D) of the dissolved substance from the measured surface concentration (2D) parameters. If the equilibrium was not reached, then using the diffusion model—the time of the interaction of the solution with the surface—it is still possible to estimate the 3D concentration of the dissolved substance based on the 2D pattern.

If the solution–surface system under study is not initially known, it is possible to draw some conclusions from the AFM image. The characteristics of particle distribution on the surface are able to be represented as a result of the process of particle flow deposition from the solution (3D) to the surface (2D). If a 3D particle approaches the surface in the vicinity of an already 2D landing particle, then, if there is an interaction between them, the deposited particle will change its trajectory toward the surface depending on the sign of the forces. In the case of attraction, the particle will sit close to the surface; in the case of repulsion, it will sit at a distance. Analyzing the distribution of particles by area makes it possible to determine the sign of the interaction force. For the colloidal systems studied in the present work, it was shown that, in the case of AuNP particles, repulsion can be observed, and for the peroxidase solution, long-range forces cannot observed.

## 4. Materials and Methods

### 4.1. Reagents

Tetrachloroauric(III) acid trihydrate, ACS reagent (Acros Organics, Fair Lawn, NJ, USA), Trisodium citrate 5.5-hydrate (“Spektr-Khim” LLC, St. Petersburg, Russia).

Lyophilized HRP powder from horseradish was purchased from Sigma (Cat. #6782, Louis, MO, USA). Dulbecco’s modified phosphate-buffered saline (PBSD) was prepared by dissolving a salt mixture, commercially available from Pierce (Waltham, MA, USA), in ultrapure water. All solutions used in our experiments were prepared using deionized ultrapure water (with 18.2 MΩ × cm resistivity) obtained with a Simplicity UV system (Millipore, Molsheim, France). A 0.1 µM solution of HRP was prepared by dissolving the above-mentioned commercially available HRP preparation in 2 mM of PBSD buffer.

### 4.2. AuNP Synthesis

To 24 mL of deionized water preheated in a shaker to ~85 °C, 0.25 mL of a 1% (38.8 mM) aqueous solution of tetrachloroauric(III) acid trihydrate under 750 rpm stirring was added. Next, 0.75 mL of a 1% aqueous trisodium citrate 5.5-hydrate solution was rapidly added. The mixture was stirred in a shaker for 30 min. The heating was then turned off, and the mixture was left to cool to room temperature with stirring. The solution was stored in the dark at 4 °C.

### 4.3. Spectrophotometric Measurements

The spectrophotometric measurements were performed in the following way. Firstly, in order to determine the size of AuNPs in the suspension, 900 µL of the initial aqueous AuNP suspension was pipetted into a 3 mL quartz cuvette (optical pathlength 1 cm; Agilent Deutschland GmbH, Waldbronn, Germany) containing 2100 µL of ultrapure deionized water and stirred thoroughly. The absorbance spectrum of the resulting suspension was recorded with an Agilent 8453 UV–vis spectrophotometer (Agilent Deutschland GmbH, Waldbronn, Germany). Secondly, in order to find out whether dilution of the AuNP suspension induces aggregation of the AuNPs, a 10-times lower volume (90 µL) of the initial aqueous AuNP suspension was pipetted into a cuvette containing 2910 µL of ultrapure deionized water and stirred thoroughly, and the absorbance spectrum of the diluted suspension was recorded. All spectrophotometric measurements were performed at room temperature in at least three technical replicates.

### 4.4. Preparation of Samples for AFM Study Adsorption of AuNPs onto a Surface

A AuNP working solution for AFM-analysis was prepared by tenfold dilution of AuNP stock solution with deionized water. A 2 μL drop of AuNP working solution was applied to a 7.5 × 15 mm mica surface using an automatic pipette and incubated in a Petri dish until completely dry. To avoid loss of AuNPs, washing was not performed.

### 4.5. Adsorption of Horseradish Peroxidase to the Surface

The initial 0.1 µM HRP solution in 2 mM of PBSD buffer was aliquoted into standard 1.7 mL Eppendorf-type test tubes (SSI Bio, Lodi, CA, USA). Then, 7.5 × 15 mm pieces of freshly cleaved bare mica were placed into the tubes and incubated for 10 min at 600 rpm in a Thermomixer Comfort laboratory shaker (Eppendorf, Hamburg, Germany) at 25 °C. In this way, HRP was allowed to directly adsorb onto mica during the incubation. After the incubation, each mica substrate with adsorbed HRP was rinsed with fresh ultrapure water and dried in the air.

### 4.6. AFM Data Processing

The AFM images were obtained with a titanium multimode atomic force microscope (NT-MDT, Zelenograd, Russia) equipped with NSG10 cantilevers (“TipsNano”, Zelenograd, Russia; 47–150 kHz resonant frequency, 0.35–6.1 N/m force constant). The microscope was operated in intermittent contact mode. The microscope was preliminarily calibrated using TGZ1 calibration grating (NT-MDT, Zelenograd, Russia; step height 21.4 ± 1.5 nm). The atomic force microscope was operated with the standard NOVA Px 3.5.0 software (NT-MDT, Moscow, Zelenograd, Russia) supplied with the microscope.

For each substrate incubated in the AuNP or HRP solution, the number of frames was ≥10, and the number of objects imaged in each sample was ≥500.

In order to check whether the noise resulting from the unspecific adsorption from the buffer solution was within its normal range (which typically makes up 500 objects per 400 µm^2^), control experiments with the use of pure buffer instead of HRP enzyme solution were performed. No objects with a height >0.5 nm were observed on the mica surface in the control experiments.

AFM images were processed by Gwyddion [35]. A Python script was written to calculate the RDF, available at https://github.com/digital-ribosome/RDF (accessed on 20 December 2024). To calculate the surface density distribution, the AFM image was divided into equal-area regions so that, on average, there were more than 25 objects in each area. A histogram was then constructed to show the probability of detecting a given number of particles in a selected area. This histogram was compared to the Poisson distribution.

## 5. Conclusions

The present work describes the rationale and results of the experimental confirmation of the possibility of using AFM data to analyze the nature of particle interactions in solution. The results for two colloidal systems with objects of an organic nature (protein molecules) and an inorganic nature (AuNPs) are presented. Processing AFM images and the construction of radial distribution and density functions for the objects allowed us to find the presence of a short-range structure in the case of AuNPs and the absence of such a structure in the case of proteins. In the case of proteins, the distribution of objects on the surface obeys the Poisson distribution; hence, the estimation of the number of objects on a limited area provides us with an idea about the objects on the whole surface with the sorbed protein.

The particle–particle and particle–surface interactions proved in the present study affect the interpretation of AFM studies, and this should be taken into account. AFM imaging results are useful in assuming the mechanism of adsorption of objects from a solution. It should be emphasized that scanning data in air were used, which is much simpler than scanning in a liquid medium. Further research is possible in applying this approach to other protein systems with different isoelectric points, as well as mixtures of colloidal particles with varying physicochemical properties. The parameters determined for these systems may help reconstruct multi-object biological systems. The results are useful to researchers in biotechnology and biosensor technologies, where systems of objects immobilized on surfaces are popular.

## Figures and Tables

**Figure 1 ijms-26-00210-f001:**
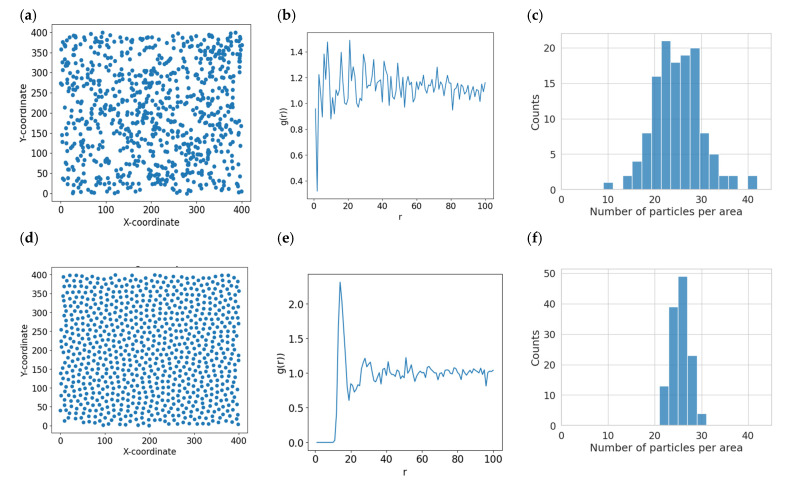
Modeling of a system of randomly arranged particles (**a**–**c**) and particles with repulsive potential (**d**–**f**). Two dimensional (2D) images (**a**,**d**); radial distribution functions (**b**,**e**); and density distribution (**c**,**f**) corresponding to the images, describing the location of the particles on the surface. LAMMPS was used for modeling single-type LJ particles with Coulomb repulsion in a 2D box with periodic boundary conditions.

**Figure 2 ijms-26-00210-f002:**
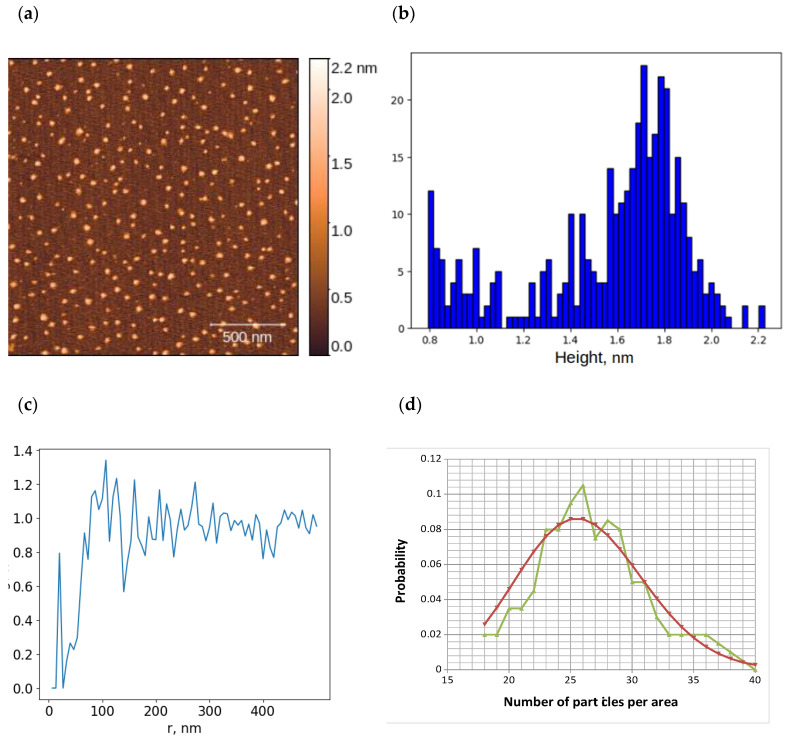
Results of atomic force microscopy (AFM) visualization of peroxidase (HRP) on the mica surface and data processing: typical AFM image of mica surface after incubation in 0.1 μM HRP solution (**a**); histogram of height distribution of visualized objects on the surface (**b**); radial distribution function for objects on the surface (**c**); experimental distribution of HRP molecules (green dots) and Poisson distribution with mean value of 26 (red dots) (**d**).

**Figure 3 ijms-26-00210-f003:**
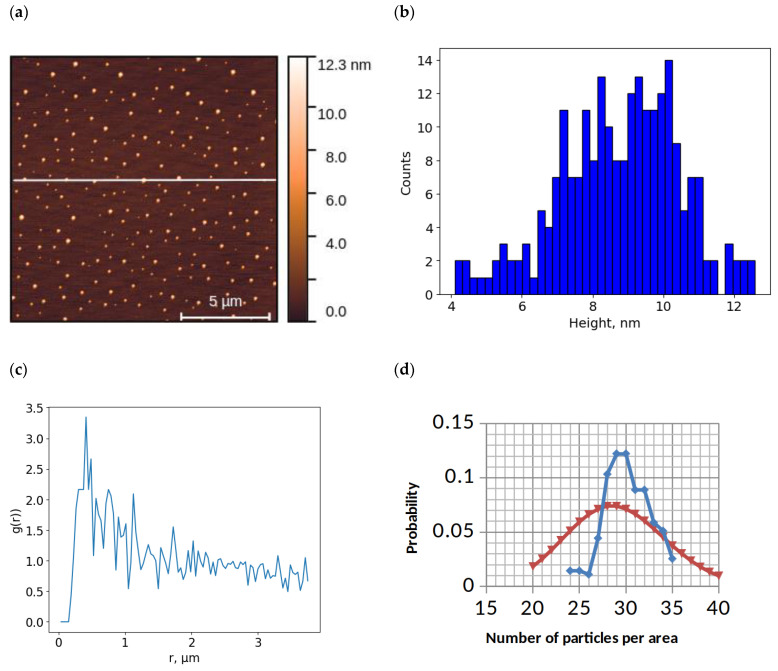
Results of AFM visualization of aurum nanoparticles (AuNPs) on the mica surface and data processing: typical AFM image of mica surface with adsorbed objects (**a**); histogram of height distribution of visualized objects on the surface (**b**); radial distribution function for objects on the surface (**c**); experimental distribution of AuNP (blue dots) and Poisson distribution with a mean value of 29 (red dots) (**d**).

**Figure 4 ijms-26-00210-f004:**
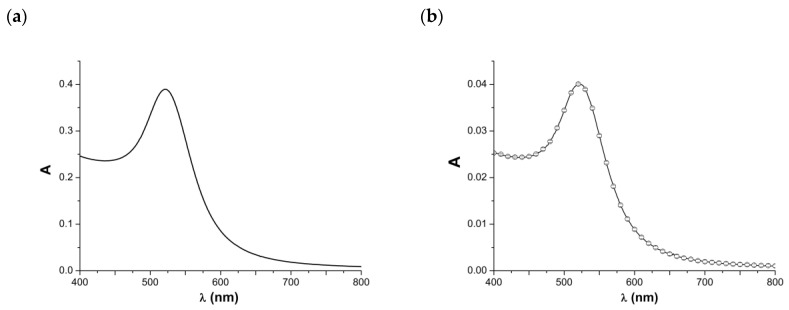
Absorbance spectrum of AuNP suspension in deionized water. Volumes of AuNP stock suspension and deionized water are 900 µL and 2100 µL, respectively (**a**); volumes of AuNP stock suspension and deionized water are 90 µL and 2910 µL, respectively (**b**).

## Data Availability

The data underlying this research can be obtained from the corresponding author upon reasonable request.

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
