# Peer review of "Using the Radial Distribution Function to Analyze Atomic Force Microscopy Images of Colloidal Systems"

_ijms, 2024, doi:10.3390/ijms26010210_

Round 1
Reviewer 1 Report
Comments and Suggestions for Authors
The manuscript titled “Using the radial distribution function to analyze AFM images of colloidal systems” by Kraevsky, S.V.; et al. is a scientific work where the authors assessed the biomolecular immobilization processes taken place on freshly cleaved mica surfaces. This study is interesting overall for those experts in this field and can open new avenues to better understand the underlying processes driven during these biomolecular immobilization. Furthermore, the manuscript is generally well-written. However, it exists some points that need to be addressed (please, see them below detailed point-by-point) to improve the scientific quality of the submitted manuscript paper before this article will be consider for its publication in the International Journal of Molecular Sciences.
1) The author should consider to add the term “colloidal particles” in the keyword list.
2) “Immobilization methods can provide both reversible binding to the surface (…) under investigation” (lines 41-44). Here, some relevant references about the biomolecular immobilization strategies [1] and how AFM can reveal the proper orientation of these biomolecules towards their respective receptors [2] should be provided and further discussed.
[1] Breitenstein, M.; et al. Immobilization of different biomolecules by atomic force microscopy. J. Nanobiotechnology 2010, 8, 10. https://doi.org/10.1186/1477-3155-8-10
[2] Marcuello, C.; et al. An efficient method for enzyme immobilization evidenced by atomic force microscopy. Protein Eng. Des. Sel. 2012, 25, 715-723. https://doi.org/10.1093/protein/gzs086
3) “Biomacromolecules generally exist and function in aqueous media” (line 45). Generally or always? Otherwise the authors should mention those cases where the biomolecules can exert their function under non-aqueous media.
4) “So for the immobilization of negatively charged DNA molecules on mica, divalent magnesium cations (…) successfully” (lines 95-97). The potential alternative of mica surfaces grafted with positive poly-L-lysine should be also discussed.
5) Figure 1 (line138). Could the authors quantify the repulsive potential extension? Then, the Gaussian fitting should be also furnished in this figure with the respective regression coefficient. Same comment for the Figure 2 (line 240).
6) Did the authors take into account the Brownian motion of biomolecules under liquid media to make the respective deposition calculations? This information is briefly discussed in the manuscript but not referred in the equations used in this research.
7) “Figure 2 shows an example of the AFM image of objects on the mica surface after incubation with HRP solution (…)” (lines 226-227). What is the isoelectrical point of HRP enzyme at the PH working conditions? This information needs to be detailed. Then, did the authors carry out a negative control with a non-charged protein like bovine serum albumin (BSA)?
8) “Conclusions” (lines 417-433). This section perfectly remarks the most relevant outcomes found by the author in this work and the promising future perspectives and some limitations to be overcome. It should be desirable to add a brief statement to discuss about the potential future action lines to pursue the topic covered in this research.
9) “4.4. Preparation of samples for AFM study adsorption of AuNPs onto the surface” (line 383-387). Why did the authors not devote any gently washing step to remove those loosely attached gold nanoparticles? Could this negatively interfere during the data acquisition? Some insights should be provided in this regard.
10) References. The number of bibliography citations (only 22) is short taken into account the field covered by the authors. This number should be enlarged.
Reviewer 2 Report
Comments and Suggestions for Authors
Dear authors and editor,
The article Using the radial distribution function to analyze AFM images of colloidal systems is 12 pages long, shorter than an average, with 12 references including among other basic books and the recently published articles.
From the introduction, it should be apparent which of data and results that the authors aim to obtain are unavailable for any system (or any system at least comparable with the system studied by the authors), or the comparable data for comparable systems reported in the literature should be collected and the newly acquired data compared with them. It is possible that it is stated in the article, but I failed in detecting it, not having too much time to deal with the article.
The Figure 1 presents some data, but the system is not specified neither by material characteristics, nor by the list of all input parameters and their values for purely theoretical modelling (or, some parameters values are in the text, but it should be clearly expressed in the legend or near the figure in the condensed form which parameters are used and their values for each case).
In Figure (2), four parts (A), (B), (C), (D) are reported, but only two graphs are presented. In the (C) part, units are missing and the quantities symbols are not cursive as they should be. The missing units for quantities that are not dimensionless and two small description are also problem of other figures.
Numerous references from the text to its supplementary material significantly lower the comfort of reading. If this information should be referred, I rate incorporating of the supplement text into the main text as much better.
To conclude, the article presents in some sections something what is not clearly described and the reader is required laboriously search the text whether the relevant information is mention somewhere, or not.
Formal remarks:
Gwyddion mentioned in line 411 (or its documentation) should be one item in the reference list.
The journal author instructions https://www.mdpi.com/journal/ijms/instructions state among other: Acronyms/Abbreviations/Initialisms should be defined the first time they appear in each of three sections: the abstract; the main text; the first figure or table. When defined for the first time, the acronym/abbreviation/initialism should be added in parentheses after the written-out form.
No exception is specified even for well known abbreviations like 2D or 3D used in the abstract (lines 12 and 13) and in the text (first time lines 46-47) without the definition.
T2 in line 165 used also without definition.
σ in line 182 used also without definition.
Meeting of this requirement is complicated by placing the Materials and Methods section (where standardly abbreviations of materials are defined) after results and discussion. It is according to the order in the author instructions, but significantly reader-unfriendly.
The same author instructions state: SI Units (International System of Units) should be used. Have the authors the special reason to use Da
The authors do not observer rules for writing symbols of variables and physical quantities.
The IUPAP recommendation is available for example at https://iupap.org/wp-content/uploads/2021/03/A4.pdf.
The IUPAC recommended documents are Quantities, Units and Symbols in Physical Chemistry (https://iupac.org/what-we-do/books/greenbook/)
and On the use of italic and roman fonts for symbols in scientific text ( https://iupac.org/wp-content/uploads/2016/01/ICTNS-On-the-use-of-italic-and-roman-fonts-for-symbols-in-scientific-text.pdf)
The International Bureau of Weights and Measures Brochure is available at https://www.bipm.org/documents/20126/41483022/SI-Brochure-9.pdf
For example, at least n and Poisson number e in lines 161-162 should be cursive. T in T2 in lines 165, 182 and other should be probably cursive as well as σ in line 182.
In chemical formulae, the numbers of atoms in the molecule should be in the subscript position. Nearly all figures break the rules of symbols writing.
For possible acceptation, the article should be rewritten following all formal rules including journal requirements, presented data and intermediate results should be specified clearly and reader friendly, and the conclusions given into the context of other published comparable results on comparable materials.
Round 2
Reviewer 1 Report
Comments and Suggestions for Authors
The authors did a great deal of work to cover all the suggestions raised by the Reviewers. For this reason, the scientific manuscript quality was greatly improved. Based on the novelty of the gathered results, I warmly endorse this work for further publication in the International Journal of Molecular Sciences.
Author Response
Thank you for comments and nice references.

Reviewer 2 Report
Comments and Suggestions for Authors
Dear authors and editor,
I have read the revised version of the article Using the radial distribution function to analyze AFM images of colloidal systems.
The authors have adequately responded the questions. I have no scientific objections. However, I have still several remarks.
First, an item that I have forgotten in the first round. I belong to people who think that abbreviations should be used in article titles. Please, think about using full "atomic force microscopy" in the article title.
Second, the authors respond "if the reviewer insists on ..." to some remarks. I believe that the editor and not the reviewer should decide these issues.
Also my remaining remarks concern the formal aspects of the article:
The authors have added abbreviations definitions into the main text, but the author instructions require that they should be defined the first time they appear in each of three sections: the abstract; the main text; the first figure or table. I think that at least "dimensional" instead bare D should be used in the abstract.
In Figure 4, the units of the horizontal axis are in cyrillic; please, replace them. I suggest to increase font size in this picture, while the labels would be sufficient for whole hundreds.
In the text, references are mostly separated by space from the previous word. Please, do it everywhere including (e.g.) line 334. The reference in line 367 given directly per DOI should be made through the references list.
The article brings some new knowledge worth of sharing, I currently do not object against acceptation. However, there are some issues in the article (some of them pointed out in the first round) that should be decided by the editor, not by reviewers who can have conflicting opinions. In addition, my subjective feeling is that the article is still written less reader-friendly than the average is.
Sincerely,
